# Dexamethasone and lactoferrin induced PMN-MDSCs relieved inflammatory adverse events of anti-cancer therapy without tumor promotion

Xing Li [1,4✉], Jie Chen[1,4], Yong-Jian Chen[1,4], Yi-Dan Qiao[1], Li-Yun Zhao[1], Nan Jiang[2], Xiang-Yuan Wu [1✉] & Yan-Fang Xing [3✉]

In this era of immune checkpoint inhibitors, inflammatory adverse events of anti-cancer therapies continue to pose a major challenge. Glucocorticoids, as the mainstay, were limited by serious side effects. Glucocorticoids induce myeloid-derived suppressor cells (MDSCs), and lactoferrin-induced polymorphonuclear MDSCs (PMN-MDSCs) were shown to relieve inflammatory conditions. Combined treatment with dexamethasone (DXM) and lactoferrin increased the generation of PMN-MDSCs in vitro (DXM/lactoferrin PMN-MDSCs) compared to DXM or lactoferrin treatment alone. DXM/lactoferrin PMN-MDSCs were distinct from tumor PMN-MDSCs in vivo with regard to gene expression profiles. DXM upregulated the myeloid cell response to lactoferrin by inducing the lactoferrin receptor Lrp1. DXM/lactoferrin PMN-MDSCs presented anti-bacterial capability, increased PGE2 production, increased survival capability, and decreased tumor tissue homing. Transfer of DXM/lactoferrin PMN-MDSCs relieved cisplatin-induced acute kidney failure, bleomycin-induced interstitial pneumonia, and allergic pneumonitis effectively without promoting tumor development. Our study shows that DXM/lactoferrin PMN-MDSCs are a promising cell therapy for inflammatory adverse events of anti-cancer therapies.

[1] Department of Medical Oncology and Guangdong Key laboratory of Liver Disease Research, the Third Affiliated Hospital of Sun Yat-sen University, 600 Tianhe Road, Guangzhou 510630, PR China. [2] Department of Transplantation, the Second Affiliated Hospital of Southern University of Science and Technology and the Third People's Hospital of Shenzhen, 29th Bulan Road, Shenzhen 510623, PR China. [3] Department of Nephrology, The Third Affiliated Hospital of Guangzhou Medical University, 63 Duobao Road, Guangzhou 510150, PR China. [4]These authors contributed equally: Xing Li, Jie Chen, Yong-Jian Chen. ✉email: lixing9@mail.sysu.edu.cn; wuxinagy@mail.sysu.edu.cn; 2574410512@qq.com

The adverse effects of chemotherapy limit the quality of life and survival of cancer patients. Among them, kidney failure[1], interstitial pneumonia[2], and allergic pneumonitis[3] feature an over-activated immune system and inflammation, and often require permanent cessation of the ongoing therapy, including cisplatin, bleomycin, and immune checkpoint inhibitors (ICIs)[4]. Glucocorticoids are considered as fundamental choice of drugs for the prevention and treatment for such inflammatory adverse events[1,5,6]. However, the adverse side effects of glucocorticoids, including opportunistic infections[7], osteonecrosis of femoral head[8], and secondary diabetes[9], makes their continued use in these cases more complicated[8,10].

Glucocorticoid was reported to induce myeloid-derived suppressor cell (MDSC) in vivo[11] and in vitro[12], which may be a major mechanism of the immuno-suppressive function of glucocorticoid. MDSCs, which are major immuno-suppressive cells, were reported to prevent early stage of doxorubicin-induced renal injury[1] and multiple inflammatory diseases[13–15] in tumor-free mice. However, in cancer patients, MDSCs induced immune tolerance to tumor antigens and promoted the development of cancer[16,17], which impedes their use in controlling inflammatory adverse events during anti-cancer therapy. There are two MDSC subgroups, polymorphonuclear-MDSCs (PMN-MDSCs) and monocytic-MDSC (M-MDSCs)[18]. M-MDSCs proliferate and transform into tumor-associated macrophages[19]. However, PMN-MDSCs do not proliferate or transform, and die within 24–48 h[19], which makes short-time transfer of PMN-MDSC a viable option in cancer patients with over-activated inflammation.

In previous study, we found that lactoferrin-induced MDSCs in vitro were efficient in treating inflammatory disease in tumor-free neonatal mice[15], and hypothesized that the efficacy may be further enhanced by the administration of glucocorticoid. In the present study, we found that dexamethasone (DXM)- and lactoferrin-induced PMN-MDSCs presented immuno-suppressive function, antibacterial capability, improved survival, decreased tumor tissue homing, as well as a distinct gene profile from the tumor PMN-MDSCs. DXM- and lactoferrin-induced PMN-MDSCs relieved anti-cancer therapy-related inflammatory adverse events without promoting tumor progression. In vitro DXM- and lactoferrin-induced PMN-MDSCs may be a remedy for the prevention and treatment of inflammatory adverse events associated with anti-cancer therapy.

## Results

### DXM and lactoferrin-induced PMN-MDSC in vitro from adult mouse bone marrow (BM) cells.

BM cells from 6–8-week-old mice were cultured for 72 h with granulocyte-macrophage colony-stimulating factor (GM-CSF) and IL-6, lactoferrin (600 µg/ml) and DXM were added with phosphate buffer saline (PBS) as control. Treatment with DXM and lactoferrin caused substantial accumulation of PMN-MDSCs (DXM/lactoferrin PMN-MDSCs) both in percentage and absolute cell number. Lactoferrin or DXM alone did not induce PMN-MDSCs significantly (Figs. 1a, b and S1a). According to our previous report[13,15], potent antibacterial activity is a prominent feature of newborn MDSCs and lactoferrin-induced MDSCs. We assessed the antibacterial activity of DXM/lactoferrin PMN-MDSCs. $1 \times 10^5$ PBS-induced PMN-MDSCs (Con PMN-MDSCs) or DXM/lactoferrin PMN-MDSCs were incubated with $10^7$ CFUs of E. coli for 2 h, and the bacterial growth was assessed 24 h later as described. DXM- and lactoferrin-induced PMN-MDSCs suppressed E. coli proliferation, while the Con PMN-MDSCs did not (Fig. 1c).

DXM/lactoferrin PMN-MDSCs, Con PMN-MDSCs, and PMN-MDSCs from spleen of B16 tumor-bearing mice (tumor PMN-MDSCs) were collected to test their immuno-suppressive

function. First, sorted CD3+ T cells from the spleen were labeled with 5, 6-carboxyfluoresceindiacetate, succinimidylester (CFSE; 2 µM; Invitrogen), stimulated with anti-CD3–coated (5 µg/mL) plates and soluble anti-CD28 antibody (1 µg/mL; eBioscience), and cultured alone or with PMN-MDSCs at 2:1 ratios for 3 d. T cell proliferation was analyzed by flow cytometry. The results revealed that only tumor PMN-MDSCs suppressed T cell proliferation and activation. Next, CD8+ T cells were isolated from the spleen of OT-I mice, labeled with CFSE, and then mixed with splenocytes from wild type (WT) mice at a 1:4 ratio and incubated with PMN-MDSCs in the presence of cognate peptides (SIINFEKL). The results revealed that PMN-MDSCs of all three groups suppressed T cell proliferation and activation, with DXM/lactoferrin PMN-MDSCs presenting intermediate suppressive effect (Fig. 1d–f). These results indicated that DXM and lactoferrin increased the suppressive function of in vitro induced PMN-MDSCs. However, the suppressive function of DXM/lactoferrin PMN-MDSC was relative weaker than that of the tumor PMN-MDSC.

### DXM and lactoferrin-induced human PMN-MDSC in vitro from human peripheral blood mononuclear cells (PBMCs).

PBMCs cells from healthy adult donors were cultured for 72 h with GM-CSF and IL-6, lactoferrin (600 µg/ml) and DXM were added with PBS as control. Lactoferrin did not induce PMN-MDSCs significantly. DXM and lactoferrin administration increased the expansion of PMN-MDSCs more efficiently than DXM alone. (Figs. 2a, b and S1b) DXM- and lactoferrin-induced human PMN-MDSCs also suppressed E. coli proliferation, unlike the control PMN-MDSCs (Fig. 2c). Next, PBS- or DXM- and lactoferrin-induced PMN-MDSCs, and PMN-MDSCs sorted from PBMC of hepatocellular carcinoma (HCC) patients were collected to test their immuno-suppressive function. CD3+ T cells from PBMCs were labeled with CFSE (2 µM), stimulated with anti-CD3–coated plates and soluble anti-CD28 antibody, and cultured alone or with PMN-MDSCs at 2:1 ratio for 3 d. The results revealed that PMN-MDSCs of all the three groups suppressed T cell proliferation and activation, with DXM- and lactoferrin-induced PMN-MDSCs presenting intermediate suppressive effect. These results consistently indicated that DXM and lactoferrin increased the suppressive function of in vitro induced PMN-MDSCs. However, the suppressive function of DXM- and lactoferrin-induced PMN-MDSCs was relative weaker than that of the tumor PMN-MDSCs (Fig. 2d–f).

### DXM- and lactoferrin-induced PMN-MDSCs are distinct from tumor PMN-MDSCs.

Next, we compared the DXM- and lactoferrin-induced PMN-MDSCs with tumor PMN-MDSCs. Lactoferrin (5 mg/mouse i.p. daily for 5 d)[13] and DXM (1 mg/Kg i.p. daily for 5 d)[11] were administered to tumor-free mice with PBS as vehicle control. DXM- and lactoferrin-induced PMN-MDSCs in vivo presented suppressive function to T cell proliferation, which was slightly weaker than tumor PMN-MDSC (Fig. S2). The spleen PMN-MDSCs from DXM/lactoferrin group and immature myeloid cells (IMCs) from the control group were collected using flow sorting. Tumor PMN-MDSCs were collected from the spleen of B16 tumor-bearing mice when the tumor was larger than 2 cm in diameter. Whole transcriptome analysis using RNA-seq was performed on the PMN-MDSCs. DXM- and lactoferrin-induced PMN-MDSCs in vivo presented strong correlation with IMC from tumor-free mice with much less correlation with tumor PMN-MDSCs (Fig. 3a). Compared to tumor PMN-MDSCs, 740 genes were upregulated and 2395 genes were downregulated in the DXM- and lactoferrin-induced PMN-MDSCs (Fig. S3). Kyoto Encyclopedia of Genes and Genomes

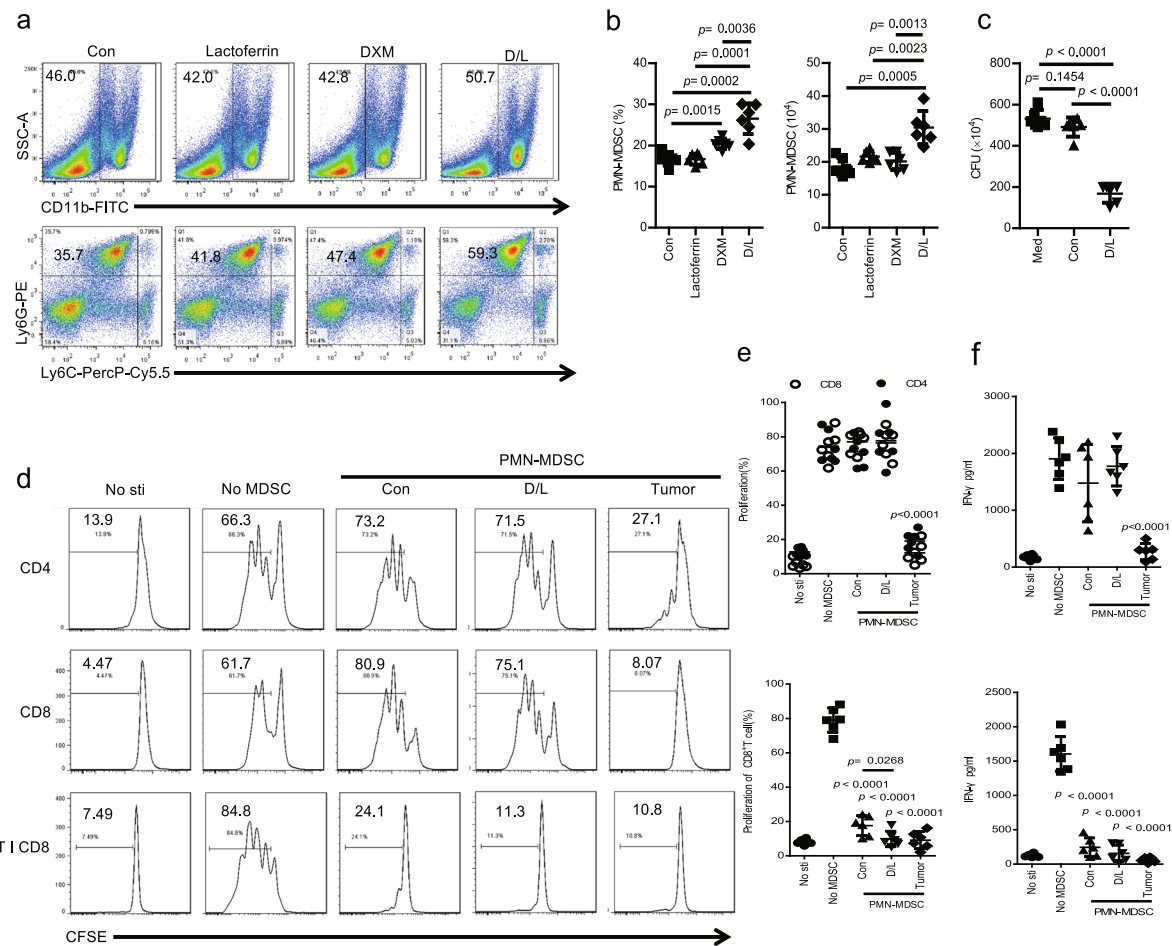

**Fig. 1 Dexamethasone (DXM)- and lactoferrin-induced PMN-MDSC in vitro from mouse bone marrow (BM) cells. a–c** BM cells from 6–8-week-old mice were treated with lactoferrin, DXM, combination of DXM and lactoferrin (D/L), or PBS vehicle (Con) in the presence of GM-CSF and IL-6 for 3 d. PMN-MDSCs were analyzed using a flow cytometer. **a** Typical example of flow cytometric analysis. **b** Absolute numbers and proportion of cells ($n = 6$). **c** Antibacterial activity of Con PMN-MDSCs and D/L PMN-MDSCs isolated from PBS- or D/L-treated mice BM cells against *E. coli* ($n = 6$ biologically independent samples). **d–f** Suppressive activity of Con PMN-MDSCs and D/L PMN-MDSCs against proliferation of T cells from WT mice stimulated with anti-CD3/anti-CD28 or OT-I–derived CD8$^+$ T cells stimulated with cognate peptide were tested by CFSE. Non-stimulated T cells (No-sti), stimulated T cells without PMN-MDSCs (No-MDSC), and tumor PMN-MDSCs from spleen of B16 bearing mice (tumor) were used as controls. **d** Typical example of flow cytometric analysis. **e** Statistical analysis. **f** IFN-γ level (in media) analyzed by ELISA ($n = 6$ biologically independent samples). Error bars displayed the mean and standard deviation.

(KEGG) pathway analysis indicated that cellular growth and death, signal transduction, lipid metabolism, and immune system were mostly different between them (Figs. S4 and 5). GO gene ontology (GO) analysis also revealed that metabolic regulation and binding function were most significantly different (Fig. S6). In vivo DXM- and lactoferrin-induced PMN-MDSCs presented higher prostaglandin E2 (PGE2)-related genes, fewer TGF-β-related genes, and altered reactive oxygen species (ROS)-related genes compared to tumor PMN-MDSCs (Fig. 3b). Thus, we speculated that DXM/lactoferrin PMN-MDSCs were different from tumor PMN-MDSCs in features such as tumor homing, survival, and immuno-suppressive functions.

**DXM upregulated myeloid cell response to lactoferrin.** Our previous study indicated that BM cells from adult mice did not respond to lactoferrin due to the lack of lactoferrin receptors[15]. RNA sequencing revealed that DXM and lactoferrin treatment increased Lrp1 expression, one of the lactoferrin receptors (Fig. 3b). Further, qRT-PCR and western blotting illustrated that Lrp1 expression was upregulated in DXM/lactoferrin PMN-MDSCs compared to Con PMN-MDSCs (Figs. 3c and S7).

Previous studies indicated that lactoferrin-induced PMN-MDSCs generated PGE2 to suppress immunity[13,15]. qRT-PCR data confirmed that PGE2 expression-related genes including Ptgs1, Ptgs2, Hpdgs, and Hpdg were relatively higher in DXM/lactoferrin PMN-MDSCs compared to tumor PMN-MDSCs in the mouse model. Consistently, Ptgs2 and Hpdgs, were relatively higher in DXM/lactoferrin PMN-MDSCs compared to tumor PMN-MDSCs from human samples (Fig. 3d). Besides, ELISA revealed that PGE2 level in the cell lysates was highest in the DXM/lactoferrin PMN-MDSCs while tumor PMN-MDSCs generated minimal amount of PGE2, both in mouse and human experiments (Fig. 3e).

**DXM/lactoferrin PMN-MDSCs presented improved survival capability, less tumor homing tendency and negative tumor promotive effects compared to tumor PMN-MDSCs.** The recovery rates were used to assess the survival capability of PMN-MDSCs[19]. DXM/lactoferrin PMN-MDSCs, Con PMN-MDSCs, and tumor PMN-MDSCs were collected and cultured without cytokines. The 24-h survival ratios (recovery rates) of the cells were highest in the DXM/lactoferrin PMN-MDSC group and

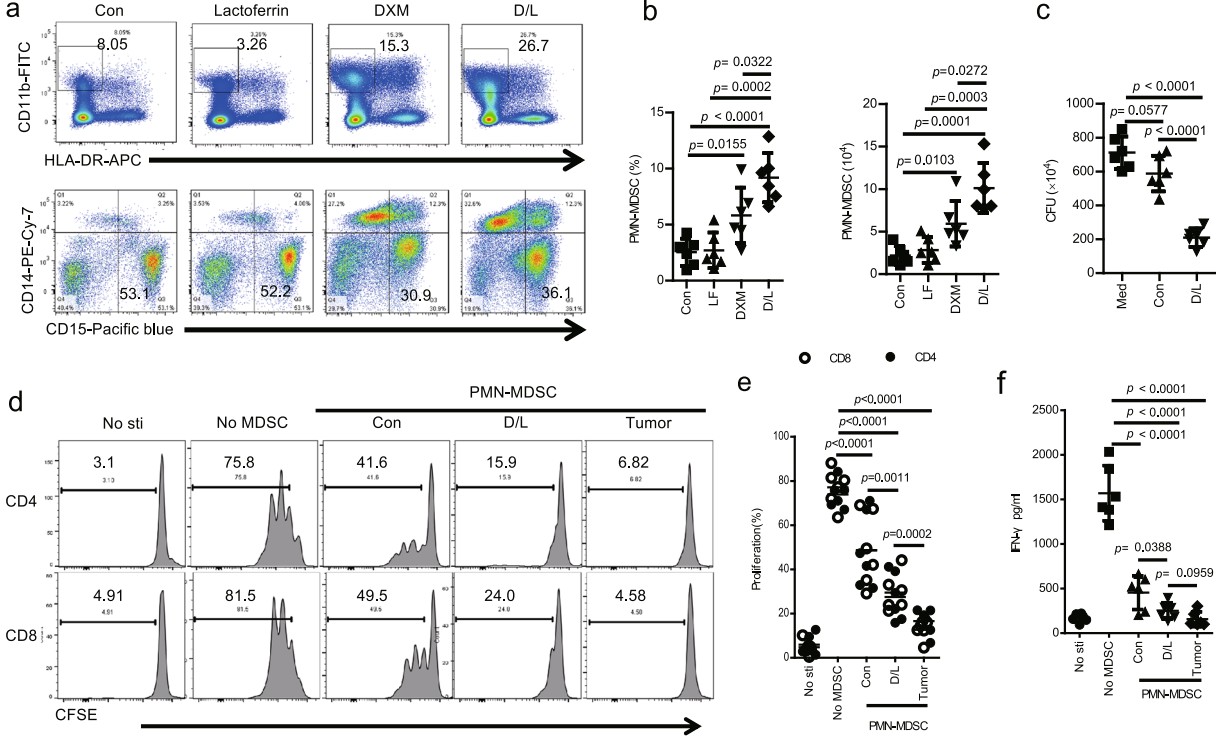

**Fig. 2 DXM- and lactoferrin-induced human PMN-MDSC in vitro from human peripheral blood mononuclear cells (PBMCs). a, b** PBMCs from an adult healthy donor were treated with lactoferrin, DXM, combination of DXM and lactoferrin (D/L), or PBS vehicle (Con) in the presence of GM-CSF and IL-6 for 3 d. PMN-MDSCs were analyzed using a flow cytometer. **a** Typical example of flow cytometric analysis. **b** Absolute numbers and proportion ($n = 6$ biologically independent samples). **c** Antibacterial activity of Con PMN-MDSCs and D/L PMN-MDSCs isolated from PBS- or D/L-treated PBMCs against *E. coli* ($n = 6$ biologically independent samples). **d–f** Suppressive activity of Con PMN-MDSCs and D/L PMN-MDSCs against proliferation of T cells from PBMC stimulated with anti-CD3/anti-CD28 were tested by CFSE. Non-stimulated T cells (No-sti), stimulated T cells without PMN-MDSC (No-MDSC), and tumor PMN-MDSCs from hepatocellular carcinoma patients were used as controls. **d** Typical example of flow cytometric analysis. **e** Statistical analysis. **f** IFN-γ level (in media) analyzed by ELISA ($n = 6$ biologically independent samples). Error bars displayed the mean and standard deviation.

lowest in the tumor PMN-MDSC group. After 36-h of culture, most of the tumor PMN-MDSC were dead while ~30% of the DXM/ lactoferrin PMN-MDSCs were still viable. All the PMN-MDSCs died after 48 h of culture (Fig. 4a).

To analyze the tumor homing and in vivo survival ability, DXM/lactoferrin PMN-MDSCs, Con PMN-MDSCs, and tumor PMN-MDSCs were labeled with CFSE (2 μM; Fig. S8) and injected intravenously (i.v.) into B16 tumor-bearing mice, when the tumors were between 0.9 and 1.1 cm in diameter. CFSE positive cells among PMN-MDSCs in spleen, PBMCs, and tumor tissues were considered transferred PMN-MDSCs. One day after the transfer, DXM/lactoferrin PMN-MDSCs and Con PMN-MDSCs accumulated in the spleen and PBMCs, while the tumor PMN-MDSCs accumulated significantly higher in the tumor tissues (Figs. 4b and S9a). After 2–3 d, the percentage of CFSE positive cells among the PMN-MDSCs in the spleen and PBMC decreased to the same level among the three groups. However, the tumor PMN-MDSCs still accumulated in the tumor tissues compared to the other two groups (Figs. 4c, d and S9b, c).

To analyze the effect of DXM/lactoferrin-induced PMN-MDSCs on both systemic and local antitumor immune responses, IFN-γ+ CD8 T cells in PBMC and tumor tissue were tested by flow cytometers 24 h after transfer of DXM/lactoferrin PMN-MDSCs, Con PMN-MDSCs, or tumor PMN-MDSCs into B16 tumor-bearing mice. PBS was used as vehicle control. In PBMC, compared with PBS group, the IFN-γ+ cells in CD8 T cells were decreased after PMN-MDSC transfer, with lowest level in tumor PMN-MDSC transferred group. However, in tumor tissues the IFN-γ+ cells in CD8 T cells were not decreased after transferring

DXM/lactoferrin PMN-MDSCs or Con PMN-MDSCs. Only tumor PMN-MDSC transfer reduced the IFN-γ+ cells in CD8 T cells in tumor tissues. (Fig. 4e, and Figs. S1c and S10).

Orthotopic mouse model of spontaneous breast cancer metastasis was utilized to evaluate the tumor promotion capacity of DXM/lactoferrin PMN-MDSCs. DXM/lactoferrin PMN-MDSCs, Con PMN-MDSCs, or tumor PMN-MDSCs were transferred of into 4T1 tumor-bearing mice three times every other day, when the subcutaneous tumors were between 0.9 and 1.1 cm in diameter. PBS was used as vehicle control. Mice were tested 1 week after transfer. transfer of tumor PMN-MDSCs and Con PMN-MDSCs promoted the growth of tumor significantly instead of DXM/lactoferrin PMN-MDSCs (Fig. 4f). Lung nodule area per mouse was highest in tumor PMN-MDSC group. Meanwhile, DXM/lactoferrin PMN-MDSCs did not promote lung metastasis (Fig. 4g, h). DXM/lactoferrin PMN-MDSCs did not promote tumor progression.

**Transfer of DXM/lactoferrin PMN-MDSCs relieved cisplatin-induced AKI without promoting tumor development.** To assess the potential application of DXM/lactoferrin PMN-MDSCs in preventing chemotherapy-associated kidney failure, cisplatin-induced acute kidney failure in B16 tumor-bearing mice was used, because malignant melanomas were not influenced by chemotherapy[20]. Adoptive transfer of DXM/lactoferrin PMN-MDSCs, Con PMN-MDSCs, or tumor PMN-MDSCs were conducted i.v. before and after cisplatin treatment, with PBS and DXM treatments serving as controls (Fig. 5a). The results revealed that transfer of DXM/lactoferrin PMN-MDSCs

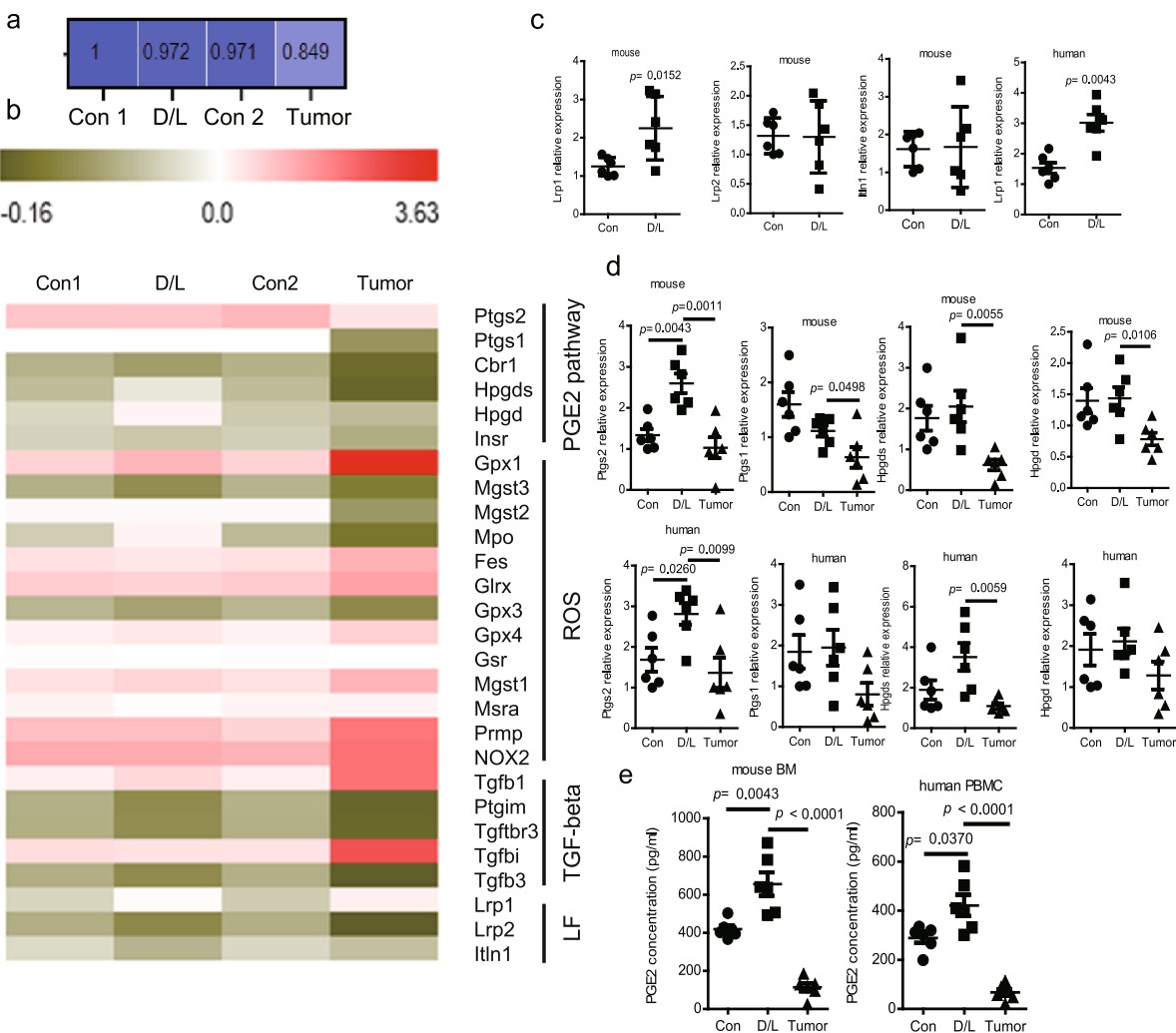

**Fig. 3 Transcriptome analysis of in vivo DXM- and lactoferrin-induced PMN-MDSC and tumor PMN-MDSCs. a** Correlation analysis of PMN-MDSC extracted from spleen of adult DXM and lactoferrin (D/L) or PBS (Con) treated tumor-free mice, and tumor-bearing mice (tumor). **b** Expression of key genes associated with the production of PGE2, ROS, and TGF-β in PMN-MDSCs of Con, D/L, and tumor groups. qRT-PCR confirmation of the gene expression of lactoferrin receptors (**c**) and PGE2-production-related genes (**d**) in PMN-MDSCs induced by PBS (Con) or D/L in vitro from mouse BM cells and human PBMCs. PMN-MDSCs from B16 tumor-bearing mice and hepatocellular carcinoma (HCC) patients were used as controls (n = 6 biologically independent samples). **e** PGE2 in PMN-MDSCs induced by PBS (Con) or D/L in vitro from mouse BM cells and human PBMCs. PMN-MDSCs from B16 tumor-bearing mice and HCC patients were used as control (n = 6 biologically independent samples). Error bars displayed the mean and standard deviation.

decreased serum creatinine, while the other treatments failed to relieve cisplatin-induced acute kidney failure (Fig. 5b). Besides, PMN-MDSCs accumulated in the kidney in PMN-MDSC transfer groups with DXM/lactoferrin PMN-MDSC group presenting highest level (Fig. 5c, d). Moreover, PMN-MDSC transfer did not promote the growth of tumor (Fig. 5e), which may be due to relative short observation period and the cisplatin-mediated elimination effect on PMN-MDSCs.

**Transfer of DXM/lactoferrin PMN-MDSCs relieved interstitial pneumonia induced by bleomycin without promoting tumor development**. The potential application of DXM/lactoferrin PMN-MDSCs in preventing bleomycin-mediated interstitial pneumonia was assessed in B16 tumor-bearing mice. Adoptive transfer of DXM/lactoferrin PMN-MDSCs, Con PMN-MDSCs, or tumor PMN-MDSCs were conducted i.v. before and after bleomycin treatment, with PBS and DXM treatments serving as controls (Fig. 6a). The results revealed that the transfer of DXM/

lactoferrin PMN-MDSCs decreased the bronchoalveolar lavage fluid (BALF) cell count (Fig. 6b), pathological grade (Fig. 6c, d) and soluble collagen in the lung (Fig. 6e), with other treatments presenting no effect. Notably, transfer of tumor PMN-MDSCs promoted the growth of tumor significantly, while Con PMN-MDSCs presented slight tumor promotion potential. However, DXM/lactoferrin PMN-MDSCs did not promote tumor progression (Fig. 6f). Moreover, PMN-MDSCs accumulated in lung in the PMN-MDSC transfer groups with DXM/lactoferrin PMN-MDSC group presenting the highest level (Fig. 6g, h).

**Transfer of DXM/lactoferrin PMN-MDSCs relieved allergic pneumonitis without promoting tumor development**. Allergic side effects were relatively higher in monoclonal antibody-based therapies, including anti-CD20 antibody, rituximab, and ICIs. Among them, allergic pneumonitis is most critical. We use OVA-induced lung inflammation in B16 mice. Adoptive transfer of DXM/lactoferrin PMN-MDSCs, Con PMN-MDSCs, or tumor

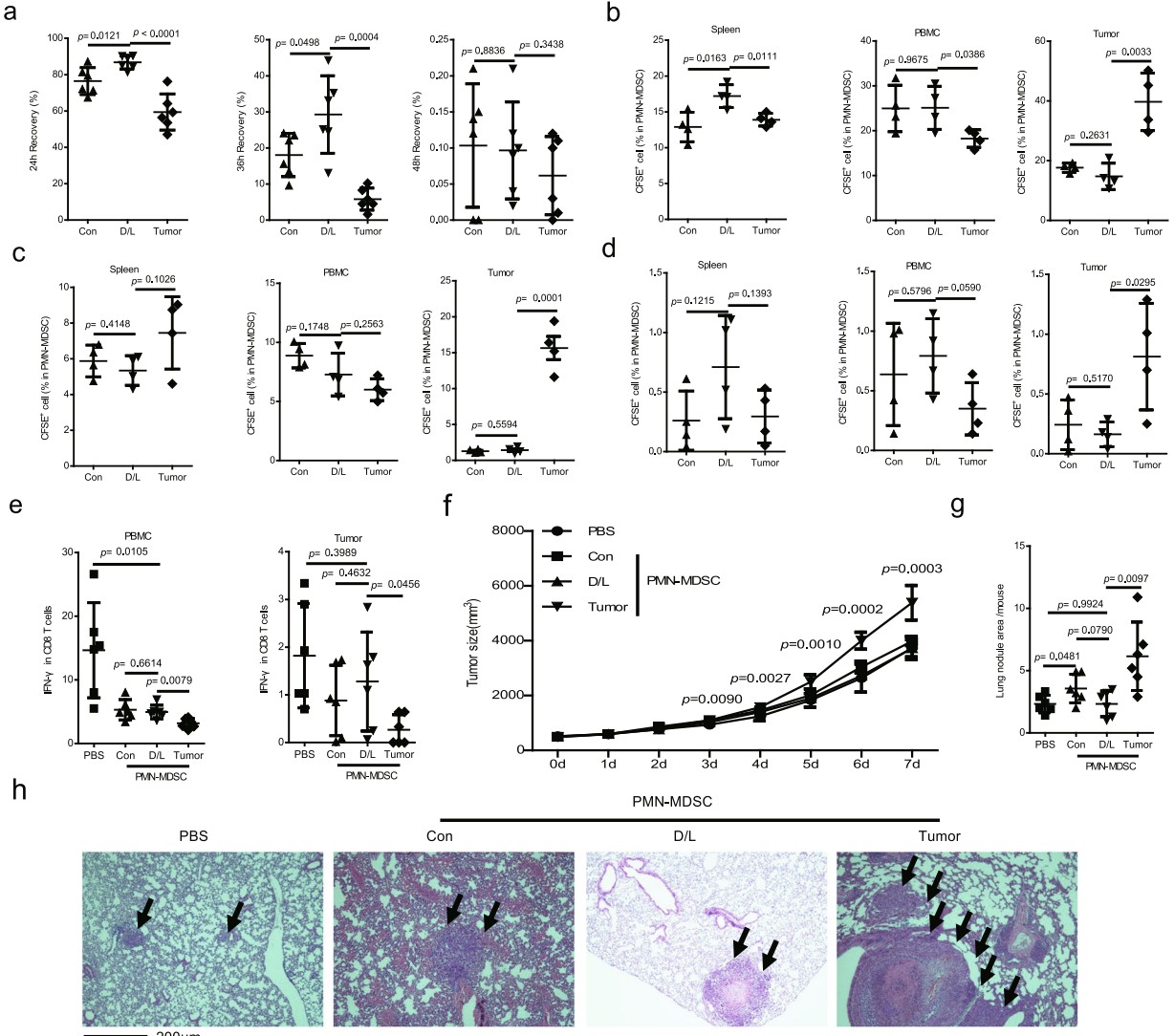

**Fig. 4 DXM/lactoferrin PMN-MDSCs presented improved survival capability, less tumor homing tendency and negative tumor promotive effects compared to tumor PMN-MDSCs. a** Recovery of PBS (Con) or DXM/lactoferrin (D/L) in vitro from mouse BM cells and PMN-MDSCs from B16 tumor-bearing mice after 24, 36, and 48 h of culture in complete media without cytokines ($n = 6$ biologically independent samples). Purified cells ($2 \times 10^5$) were plated at the beginning of culture. Transfer/host PMN-MDSC ratio in spleen, PBMC, and tumor 1 (**b**), 2 (**c**) and 3 (**d**) days after i.v. injection of CFSE-labeled PBS (Con) or D/L in vitro from mouse spleen cells and PMN-MDSCs from B16 tumor-bearing mice into tumor-bearing recipients. Percentages of CFSE-labeled PMN-MDSC are indicated ($n = 4$ biologically independent animals). **e** IFN-$\gamma^+$ cells in CD8 T cells from PBMC and tumor of B16 tumor-bearing mice 24 h after i.v. injection of Con PMN-MDSCs, D/L PMN-MDSCs, and tumor PMN-MDSCs. PBS was used as vehicle control. Tumor growth (**f**) and lung metastasis (**g**) after i.v. injection of Con PMN-MDSCs, D/L PMN-MDSCs, and tumor PMN-MDSCs. **h** Representative images of hematoxylin and eosin-stained sections of the lungs ($n = 6$ biologically independent animals). Error bars displayed the mean and standard deviation.

PMN-MDSCs were conducted i.v. during OVA challenge, with PBS and DXM treatment serving as controls (Fig. 7a). The results revealed that transfer of DXM/lactoferrin PMN-MDSCs, Con PMN-MDSCs, and tumor PMN-MDSCs transfer and treatment with DXM decreased the bronchoalveolar lavage fluid (BALF) cell count (Fig. 7b), eosinophil granulocyte percentage (Fig. 7c), pathological grade (Fig. 7d, e), and IL-4 concentration (Fig. 7f). DXM/lactoferrin PMN-MDSCs presented relatively stronger effect than Con PMN-MDSCs. Notably, transfer of tumor PMN-MDSCs transfer promoted the growth of tumor significantly, with Con PMN-MDSCs and DXM/lactoferrin PMN-MDSCs presenting no tumor promotion potential (Fig. 7g). PMN-MDSC accumulated in lung in all the three PMN-MDSC transfer groups with DXM/lactoferrin PMN-MDSC group presenting the highest level (Fig. 7h, i).

## Discussion

Inflammatory adverse events during anti-cancer therapy are characterized by inflammatory infiltration and reduction in function of target organs, especially the lung and kidneys[1–5]. Acute kidney failure and pneumonia not only delay the anti-cancer therapy for uncertain periods, but also lead to dose reduction or cessation of the effective treatment. In this era of ICIs, undesirable impairment of immunotolerance of non-tumoral tissues has become more frequent, and is classified as immune-related adverse events (irAE)[4,5]. The adverse effects of such anti-cancer treatments are similar in nature to those occurring in autoimmune diseases. MDSCs were reported to be effective in the treatment of inflammatory diseases including inflammatory bowel disease (IBD)[21], allergic asthma[14,22], necrotizing enterocolitis (NEC)[13,15], abortion[23], ConA-induced

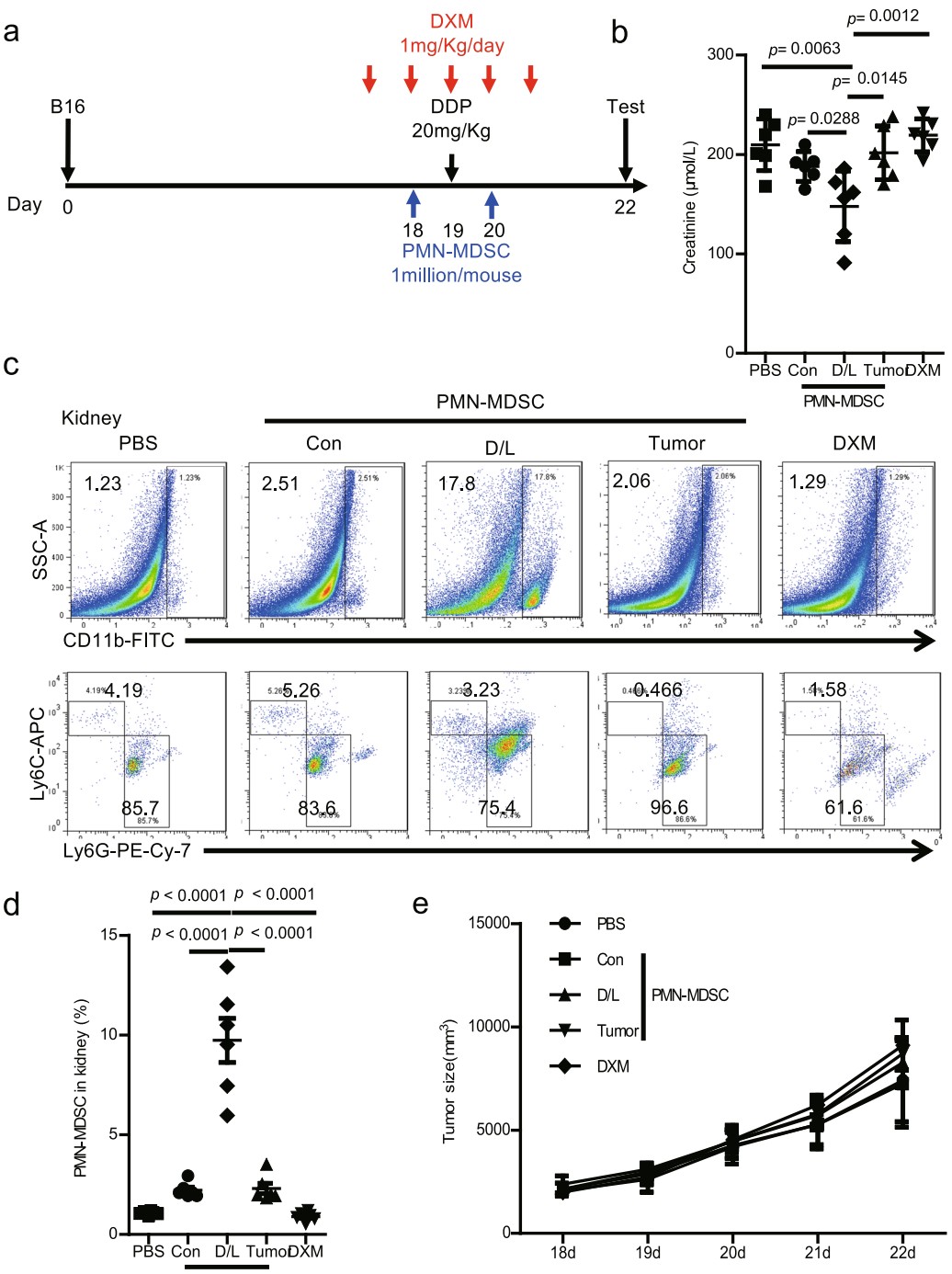

**Fig. 5 Transfer of DXM/lactoferrin PMN-MDSCs relieved cisplatin-induced AKI without promoting tumor development. a** Diagram of the experimental design. DDP (20 mg/Kg) was administered intratracheally to wild type (WT) mice 19 d after tumor implantation, and tested on day 22. Cells (1 × 10[6]) of Con PMN-MDSC, DXM/lactoferrin (D/L) PMN-MDSC, tumor PMN-MDSC, or vehicle control (PBS) was administered i.v. on days 18 and 20. DXM was administered at 1 mg/kg/day from day 17 through 21 in the indicated group. **b** Serum creatinine level in each group. **c** Typical example of flow cytometric analysis and **d** statistical analysis of PMN-MDSCs in the mouse kidneys. **e** Tumor growth (n = 6 biologically independent animals). Error bars displayed the mean and standard deviation.

hepatitis[15], and chemotherapy-mediated kidney injury[1]. However, the low induction efficiency in vitro, affecting host anti-infection immunity[24], and potential of promoting cancer[16,17] has hindered the clinical applicability of MDSCs in the prevention and treatment of inflammatory adverse events due to anti-cancer therapy. The present study found that DXM- and lactoferrin-induced PMN-MDSCs relieved anti-cancer therapy-related adverse events in the kidneys and lungs without affecting anti-infection immunity or promoting cancer development.

Previous studies found that lactoferrin-induced MDSCs from newborn mice BM cells and not from adult BM cells, because of the substantial reduction of expression of the lactoferrin receptor

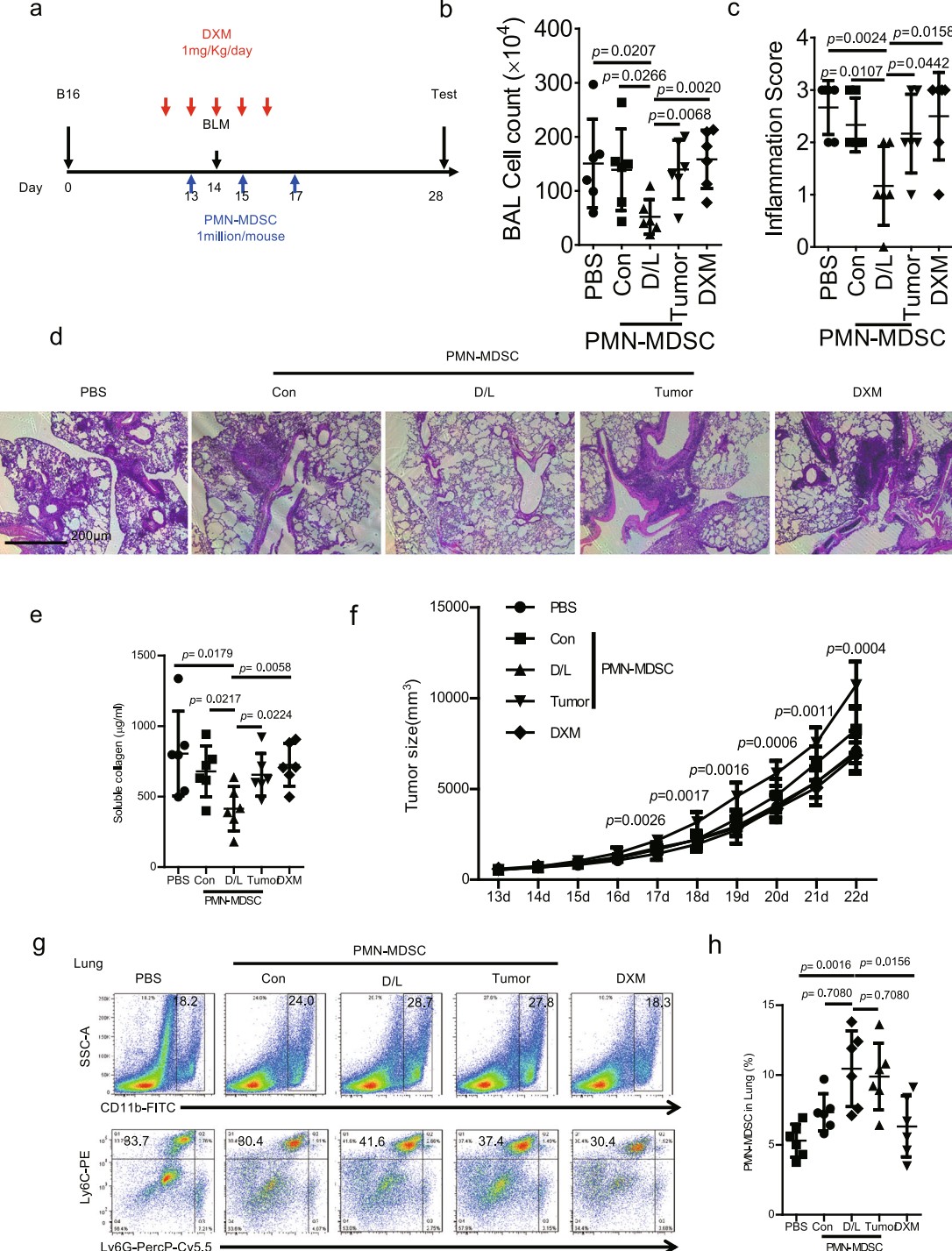

**Fig. 6 Transfer of DXM/lactoferrin PMN-MDSC relieved bleomycin-induced interstitial pneumonia without promoting tumor development. a** Diagram of the experimental design. BLM sulfate (2 U/kg) was administered intratracheally to wild type (WT) mice through tracheotomy under isoflurane anesthesia 14 d after tumor implantation, and tested on day 28. For PMN-MDSC transfer experiments, $1 \times 10^6$ cells of Con PMN-MDSC, DXM/lactoferrin (D/L) PMN-MDSC, tumor PMN-MDSC, or vehicle control (PBS) was administered i.v. before BLM challenge on day 13 every other day three times. DXM was administered at 1 mg/kg/day from day 12 through day 16 in the indicated group. **b** Total cell counts in the bronchoalveolar lavage fluid (BALF). **c** Histopathological scores. **d** Representative images of hematoxylin and eosin-stained sections of the lungs. **e** Collagen content in the lungs. **f** Tumor growth. **g** Typical example of flow cytometric analysis and **h** statistical analysis of PMN-MDSCs in the lungs ($n = 6$ biologically independent animals). Error bars displayed the mean and standard deviation.

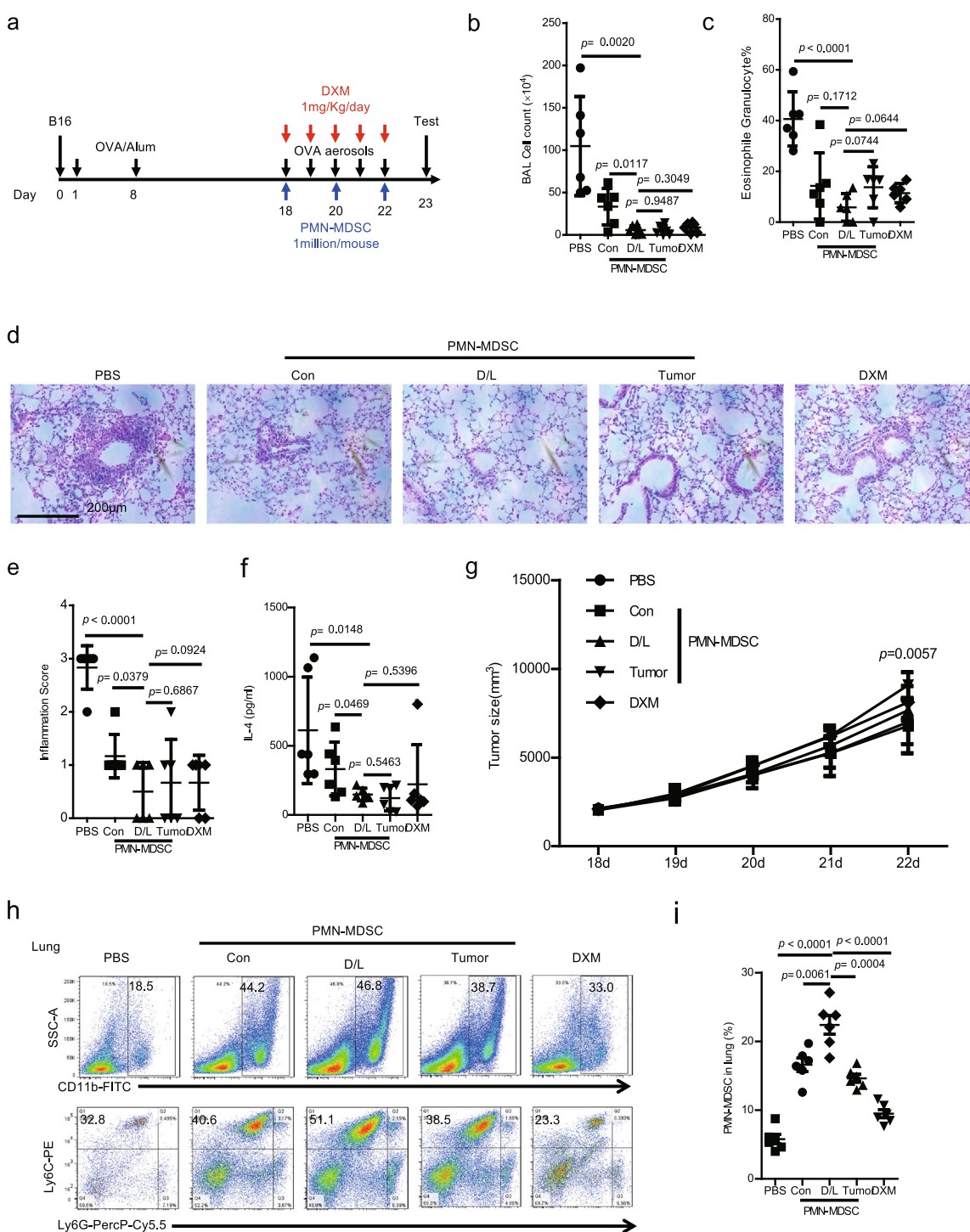

**Fig. 7 Transfer of DXM- and lactoferrin-induced PMN-MDSC relieved allergic pneumonitis without promoting tumor development. a** Diagram of the experimental design. Adult mice were intraperitoneally sensitized with 5 μg/g OVA emulsified in 200 μg/g of aluminum hydroxide on days 1 and 8 after tumor cell implantation, followed by intranasal challenge with OVA (5 μg/μl of PBS) daily on day 18 through 22. One day after the last challenge, the mice were sacrificed, and the lung inflammation was examined. For PMN-MDSC transfer experiments, 1 × 10$^6$ cells of Con PMN-MDSC, DXM/lactoferrin (D/L) PMN-MDSC, tumor PMN-MDSC, or vehicle control (PBS) were administered i.v. before the first challenge every other day three times. DXM was administered at 1 mg/kg/day followed by every challenge in the indicated group. **b** Total cell counts in the bronchoalveolar lavage fluid (BALF). **c** Eosinophil granulocyte percentage in BALF cells. **d** Representative images of hematoxylin and eosin-stained sections of the lungs and **e** Histopathological scores. **f** IL-4 level in BALF by ELISA. **g** Tumor growth. **h** Typical example of flow cytometric analysis and **i** statistical analysis of PMN-MDSCs in the lungs (n = 6 biologically independent animals). Error bars displayed the mean and standard deviation.

on the surface of adult myeloid cell compared to neonatal myeloid cells[13,15]. The current study found that DXM induced Lrp2 expression in adult myeloid cells rendered them responsive to lactoferrin. Consequently, the combination of DXM and lactoferrin-induced PMN-MDSCs both in PBMCs of adult human and BM cells of adult mice. Besides, previous studies found that DXM mainly induced M-MDSC in vitro[12]. The present study found that DXM induced PMN-MDSCs. However, the combination of DXM and lactoferrin caused a more reliable and efficient induction of PMN-MDSCs. Thus, the present study reports an efficient strategy for the in vitro induction of PMN-MDSCs.

Immuno-suppressive cells, including PMN-MDSCs, hinder host anti-infection immunity and lead to unfavorable prognosis,[17,24,25]. This limits the clinical application of PMN-MDSCs. Lactoferrin-induced PMN-MDSCs and neonatal PMN-MDSCs were found to present anti-bacteria capability which balances the immuno-suppressive function and the risk of inducing infectious disease[13,15]. In the present study, DXM- and lactoferrin-induced PMN-MDSCs displayed antibacterial function which was mainly due to lactoferrin.

The promotion of tumor progression is the main risk associated with the clinical application of PMN-MDSCs. Transfer of PMN-MDSCs has been confirmed to promote tumor progression in many previous studies[19,26]. The mechanism of tumor progression mediated by PMN-MDSCs included immune suppressive and non-immunological ones[16,27]. Tumor PMN-MDSCs presented significantly different gene expression profiles compared to IMC[13]. However, DXM- and lactoferrin-induced PMN-MDSCs in vivo resemble IMC more than tumor PMN-MDSCs in gene profile. Besides, DXM/lactoferrin PMN-MDSCs presented less tumor tissue homing, weaker immuno-suppression, and negative influence to antitumor immune response in tumor tissues. As a result, transfer of DXM/lactoferrin PMN-MDSCs, instead of Con PMN-MDSC, did not promote tumor metastasis and growth. Thus, DXM/lactoferrin PMN-MDSCs were different from the tumor PMN-MDSCs and presented reduced tendency to promote tumor progression. However, DXM/lactoferrin PMN-MDSCs transfer in this study was no more than 3 times or 5 days. The safety of long term usage of DXM/lactoferrin PMN-MDSCs transfer and its therapeutic regimen on human was still uncertain.

The target population of DXM/lactoferrin PMN-MDSCs would be patients with autoimmune diseases and late-stage cancer with inflammatory adverse events. Patients receiving radical treatment, adjuvant therapy, and neoadjuvant therapy should not be the target population. Besides, multiple anti-cancer agents were reported to kill PMN-MDSCs, including chemotherapy drugs[27]. Thus, the optimum time for clinical use of DXM/lactoferrin PMN-MDSCs would be prior to chemotherapy. Generation of PMN-MDSCs from the PBMCs of cancer patients may lead to a small population of tumor PMN-MDSC among the DXM/lactoferrin PMN-MDSCs. In such cases, the expression of lactoferrin receptor may help in sorting out the tumor PMN-MDSCs to some extent. Further studies based on clinical samples are need for the clinical application of DXM/lactoferrin PMN-MDSCs.

Above all, the present study found that DXM- and lactoferrin-induced PMN-MDSCs presented immuno-suppressive function, antibacterial capability, improved survival, decreased tumor tissue homing, as well as a distinct gene profile from the tumor PMN-MDSCs. DXM- and lactoferrin-induced PMN-MDSCs relieved anti-cancer therapy-related inflammatory adverse events without promoting tumor progression. In vitro DXM- and lactoferrin-induced PMN-MDSCs may be a remedy for the prevention and treatment of inflammatory adverse events associated with anti-cancer therapy.

## Methods

**Ethics statement**. This study was approved by the Clinical Ethics Review Board of the Third Affiliated Hospital of Sun Yat-Sen University. A written informed consent was obtained from all the patients at the time of admission. The animal study was approved by the Institutional Animal Care and Use Committee of the Sun Yat-Sen University.

**Mouse and tumor models**. 6–8-week-old female and male C57BL/6 and BALB/C mice were purchased from the Laboratory Animal Center of the Sun Yat-Sen University and OT-1 transgenic mice were kindly provided by Hui Zhang (Sun Yat-Sen University)[13]. All mice were bred in a pathogen-free facility. To establish the tumor growth models, $1 \times 10^6$ B16-F10 tumor cells (purchased from ATCC) were injected subcutaneously (s.c.) into the flank of the mice. Tumor growth was measured every day with a caliper, and volumes were calculated as V = ½ (length (mm) × (width (mm))²)[26].

**In vitro generation of PMN-MDSCs**. Mouse BM cells were harvested by flushing the femoral bones and filtering the cell suspension through an 80 μm pore size cell strainer (Corning, NY, USA). BM cells or human PBMCs ($2 \times 10^6$/ml) were cultured in RPMI 1640 medium supplemented with 10% FBS, 20 ng/ml GM-CSF, 50 μM 2-ME, and 20 ng/ml IL-6 in 24-well plates. The cultures were maintained at 37 °C in 5% $CO_2$-humidified atmosphere. Cells were analyzed by flow cytometry on day 3. DXM (0.16 μM) and/or lactoferrin protein (600 μg/ml) were added in the indicated experiments (All reagents are listed in Table S1).

**In vivo induction of PMN-MDSCs**. For the in vivo experiments, 6–8-week-old WT mice were treated with lactoferrin (Sigma Aldrich, St. Louis, MO, USA; 5 mg/mouse i.p. in PBS daily) and DXM (1 mg/kg i.p. in PBS daily) for 5 d. PBS was used as a vehicle control. At the end of the treatment, the spleens were excised, mechanically dissociated, and filtered with ammonium-chloride-potassium (ACK) lysis buffer. The PMN-MDSCs from the indicated mice were sorted by flow cytometric sorting.

**Patients and healthy donors**. Blood samples were collected from eighteen adult healthy donors and six late-stage HCC patients between February, 2019 and July, 2019 at the Third Affiliated Hospital of Sun Yat-Sen University, Guangzhou, China. The blood samples were analyzed within 6 h after sampling. PBMCs were isolated from the whole blood by Ficoll centrifugation. The diagnosis of HCC was confirmed by pathology or using the American Association for the study of liver diseases radiological criteria by either computed tomography (CT) or magnetic resonance imaging (MRI). All patients and healthy controls were also screened for serum human immune-deficiency virus (HIV) antibody, hepatitis B surface antigen (HBsAg), hepatitis C virus (HCV) antibody, hepatitis D virus (HDV) antigen, and HDV antibody. Patients and healthy controls who tested positive for HIV, chronic hepatitis virus infection (except for hepatitis B virus in HCC patients), and other acute infections (including pneumonia, urinary tract infection, etc.), who were pregnant, who received systematic corticosteroids, immuno-suppressive agents, or anti-cancer therapies and those with fever were excluded from this study.

**Flow cytometric analysis and sorting**. Cell phenotypes were analyzed by flow cytometry using a flow cytometer, FACSAria II flow cytometer (BD Bioscience), and the data were analyzed with the FlowJo V10.0.7 (FlowJo, OR, USA). For the flow cytometric sorting, a BD FACSAria cell sorter (BD Bioscience) was used. The gating strategies for PMN-MDSC was CD11b+Ly6G$^{high}$Ly6C− in mouse and CD11b+HLA-DR$^{low/−}$CD15+CD14− in human. PBMCs of HCC patients were isolated from whole blood by Ficoll centrifugation. PMN-MDSCs were collected by flow cytometric sorting. The antibodies used in flow cytometry are listed in Table S2.

**Intracellular staining**. Mononuclear cells were induced with brefeldin A (BD Biosciences), monensin, (BD Biosciences), followed by a 4 h incubation. Then the mononuclear cells were harvested, stained with antibodies for surface markers, permeabilized using Fixation/Permeabilization Solution Kit (BD) overnight at 4 °C. After washing (PBS) the cells were incubated with additionally stained for intra-cellular cytokines. The labeled cells were analyzed by flowcytometry.

Surface staining for CD8 was performed on freshly isolated monouclear cells. Cells were then washed in PBS and fixed for 60 min at 25 °C (room temperature) using the fixation/perme-abilization kit (BD), according to manufacturer instructions. After washing (PBS) the cells were incubated with anti-mouse IFN-γ antibody at 25 °C (room temperature). After 3 additional wash steps, the labeled cells were analyzed by flowcytometry.

**PMN-MDSC suppression assay**. T cell proliferation was determined by CFSE (5, 6-carboxyfluorescediacetate, succinimidylester) dilution. Purified T cells were stained with CFSE (3 μM; Invitrogen), stimulated with 0.5 μg/ml 3-h pre-coated anti-CD3 and 0.5 μg/ml anti-CD28 (eBioscience), and cultured alone or cocultured with indicated PMN-MDSCs at different ratios for 72 h. The cells were then labeled for the expression of surface marker with CD4-PE or CD8-PE-Cy5 antibodies, and

T cell proliferation was analyzed using a flow cytometer. All cultures were carried out in the presence of 20 IU/ml recombinant human/mouse IL-2 (PeproTech, Rocky Hill, NJ) in RPMI 1640 (Life Technologies) for 72 h at 37 °C.

To evaluate the antigen-specific T cell-activation, the splenocytes from the OT-1 transgenic mice were activated in the presence of OT-1-SIINFEKL (Sigma Aldrich).

**ELISA.** Cell supernatants were collected, centrifuged at $300 \times g$ for 5 min at 4 °C, and filtered through a 0.22 μm filter. IFN-γ ELISA kit (DKW12-1000-09, Dakewe Bioengineering Co., Shenzhen, Guangdong) was used to measure IFN-γ in the conditioned media of PMN-MDSCs and T cell co-culture system. Serum creatinine ELISA kit (100-300-SCR, Alpha Diagnostic International) was used to test serum creatinine in the mice. PGE2 in cell lysates was tested by ELISA according to the manufacturer's protocol (Invitrogen, Carlsbad, CA, USA)

**RNA-sequencing data analysis.** PMN-MDSCs from the spleens of DXM- and lactoferrin-treated WT mice, tumor mice, and control mice were sorted on a FACSAria cell sorter (BD Bioscience). The sorting purity was >95%. RNA sequencing was performed using BGIseq500 platform (BGI-Shenzhen, China). Single-end read runs were used, with read lengths up to 50 bp in high output mode, 20 M total read counts. Data were aligned using RSEM v1.2.12 software against mm10 genome and gene-level read counts and RPKM values on gene-level were estimated for ensemble transcriptome. Samples with at least 80% aligned reads were analyzed. DESeq2 was used to estimate significance between any two experimental groups. Overall changes were considered significant if passed FDR < 5% thresholds were taken to generate the final gene set.

**Quantitative real-time PCR (qRT-PCR).** RNA was extracted using a Multisource Total RNA Miniprep Kit (AXYGEN, CA, USA) and qRT-PCR was performed using commercially available primers and SYBR Premix Ex Taq II (DRR081; Takara Biotechnology Co. Ltd., Dalian, China). Fluorescence for each cycle was quantitatively analyzed using the ABI Prism 7000 sequence detection system (Life Technologies). The results were reported as relative expression, normalized to β-actin housekeeping gene as an endogenous control and expressed in arbitrary units. The sequence of the primers used is listed in Table S3.

**Cisplatin-induced acute kidney failure model.** Cisplatinum (DDP; 20 mg/Kg) was administered intratracheally to 6–8-week-old WT mice 19 d following tumor implantation, and tested on day 22. For PMN-MDSC transfer experiments, $1 \times 10^6$ cells of Con PMN-MDSCs, DXM/lactoferrin PMN-MDSCs, tumor PMN-MDSCs, or vehicle control (PBS) were administered i.v. on d 18 and 20. DXM was administered at 1 mg/Kg/day from day 17 through day 21 in the indicated groups.

**BLM induced interstitial pneumonia model.** BLM sulfate (2 U/kg) was administered intratracheally to 6–8-week-old WT mice through tracheotomy under isoflurane anesthesia 14 d after tumor implantation, and tested on day 28. For PMN-MDSC transfer experiments, $1 \times 10^6$ cells of Con PMN-MDSCs, DXM/lactoferrin PMN-MDSCs, tumor PMN-MDSCs, or vehicle control (PBS) were administered i.v. before the BLM challenge on day 13 every other day three times. DXM was administered at 1 mg/Kg/day from day 12 through day 16 in the indicated groups.

Sections (5 mm) of paraffin-embedded lung tissues were stained with hematoxylin and eosin (H&E) according to standard histological procedures and optical observation of the microscopic lung structure was performed blindly by an expert pathologist. Lung inflammation was assigned a score from 0 to 3 based on the absence(0) or presence to a mild[1], moderate[2], severe[3] degree of focal thickening of alveolar membranes, congestion, and interstitial and intra-alveolar neutrophil infiltration. A total cumulative histology score was determined[2].

**Allergy-induced airway inflammation mouse model.** C57BL/6 mice (6–8 weeks old) were intraperitoneally sensitized with 5 μg/g OVA (Grade V, Sigma Aldrich) emulsified in 200 μg/g of aluminum hydroxide (ThermoImject® Alum) on days 1 and 8 after tumor cell implantation, followed by intranasal challenge with OVA (5 μg/μl in PBS) once on day 18 through day 22. One day after the last challenge, the mice were sacrificed, and lung inflammation was examined. For PMN-MDSC transfer experiments, $1 \times 10^6$ cells of Con PMN-MDSCs, DXM/lactoferrin PMN-MDSCs, tumor PMN-MDSCs, or vehicle control (PBS) were administered i.v. before the first challenge every other day three times. DXM was administered at 1 mg/Kg/day followed by the challenge in indicated group. Eosinophil granulocyte in BALF was tested using a flow cytometer and defined as CD3−CD19−Gr-1−CCR3+ (Fig. S1D). Peribronchiolar and perivascular inflammation in hematoxylin and eosin-stained slides was assessed as follows: 0, normal; 1, infrequent inflammatory cells; 2, a ring of 1 layer of inflammatory cells; 3, a ring of 2–4 layers of inflammatory cells; and 4, a ring of more than 4 layers of inflammatory cells[22].

**Mononuclear cells isolation from tissues.** Lungs, tumor tissues, and kidneys were excised, minced on ice, and then incubated with 1 mg/ml collagenase D and 100 U/ml DNaseI in RPMI 1640 medium in a volume of 15 mL per organ for 90 min (lung) or 45 min (tumor tissue and kidney) at 37 °C with continuous agitation

in an incubator. The crude suspensions were further filtered through 70-μm cell strainers to obtain single-cell suspensions. The single-cell suspensions were fractionated using 40%/80% Percoll (GE Healthcare, Uppsala, Sweden) density gradient centrifugation (lung) or Ficoll density gradient centrifugation (kidney and tumor tissue).

**Antimicrobial activity.** Con PMN-MDSCs or DXM/lactoferrin PMN-MDSCs $(1 \times 10^5)$ were incubated with $10^7$ CFUs of E. coli for 2 h, and bacterial growth was assessed 24 h later[13,15].

**Orthotopic mouse model of spontaneous breast cancer metastasis.** $1 \times 10^4$ 4T1-Luc breast cancer cells was subcutaneously (s.c.) injected into the flank of 6–8-week-old BALB/c mouse. $1 \times 10^6$ cells of Con PMN-MDSCs, DXM/lactoferrin PMN-MDSCs, tumor PMN-MDSCs, or vehicle control (PBS) were administered i.v. when the tumors were between 0.9 and 1.1 cm in diameter three times every other day. One week after transfer, mice were anesthetized under constant flow of 2% isoflurane and oxygen. The lungs were inflated with formalin, followed by nodules counting and hematoxylin/eosin staining. The micro-nodules were analyzed by ImageJ for pixel counts. The pixel numbers of the nodules were divided by total lung area, and average three sections were analyzed for each animal[26].

**Statistics and reproducibility.** For most of the experiments, statistical analyses were done using unpaired t tests. Survival curves were compared using the Log-rank (Mantel–Cox) test and Gehan–Breslow–Wilcoxon tests. $N = 6$ in most of the experiments and repeated by at least 3 times to ensure the reproductivity. Statistical tests were performed using GraphPad Prism version 7.0 and SPSS Statistics 20.0. $P$ values of 0.05 (two sided) were considered significant.

**Reporting summary.** Further information on research design is available in the Nature Research Reporting Summary linked to this article.

## Data availability

All data associated with this study are present in the paper, the Supplementary Information and the Supplementary Data. Sequences of this study were uploaded in www.ncbi.nlm.nih.gov/sra/?term=PRJNA693869[28].

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

## Acknowledgements

We are grateful to Na Cheng for providing pathological diagnosis and grading. We acknowledge Meng-Xing Wang and Gang Qin for their contributions to samples transporting work. This study was supported by Guangdong Basic and Applied Basic Research Foundation (Nos. 2019A1515012198, 2019A1515011187, 2017A030313537), Guangzhou Science and Technology Project (201904010461), National Natural Science Foundation of China (81972677, 81871999 and 81700645), Special Fundamental Research Fund of Sun Yat-sen University (19ykpy17) and Tip-top Scientific and Technical Innovative Youth Talents of Guangdong Special Support Program (2019TQ05Y266).

## Author contributions

J.C., Y.-F.X. and J.-Y.C. designed the experiments, performed most of these experiments, analyzed the data, and wrote the manuscript. L.-Y.Z. and Y.-D.Q. performed some of the experiments. N.J. carried out the bioinformatics analysis. X.L. and X.-Y.W. contributed to the idea generation, experimental design, manuscript writing, and conceived the project.

## Competing interests

The authors declare no competing interests.
