## [Peer Review File · Communications Biology]

Fig. S1. Gating strategy of the eosinophil granulocyte percentage in PMN-MDSC of mice (a) and human (b), and $\text{INF-}\gamma^+$ CD8 T cells (c) and BALF cells (d).

Fig. S2. Immune suppressive activity of DXM/lactoferrin induced PMN-MDSCs in vivo. DXM/lactoferrin (D/L) induced PMN-MDSCs in vivo were co-cultured with T cells from spleen with different ratio. T cells were labeled by CFSE and stimulated with anti-CD3/anti-CD28 and Tumor PMN-MDSCs from hepatocellular carcinoma patients were used as controls. T cell proliferation were tested by flow cytometer and INF- γ in conditioned media tested by ELISA. (a) Typical example of flow cytometric analysis. Statistical analysis of T cell proliferation (b) and INF- γ (c). (n=4 biologically independent samples) Error bars displayed the mean and standard deviation.

Fig. S3. Analysis of differentially expressed genes in dexamethasone and lactoferrin *in vivo* induced PMN-MDSCs vs. tumor PMN-MDSCs by volcano plot analysis.

KEGG-pathway classification

Fig. S4. KEGG-pathway classification of dexamethasone and lactoferrin *in vivo* induced PMN-MDSCs compared to tumor PMN-MDSCs

KEGG-pathway enrichment chart

Fig. S5. KEGG-pathway enrichment chart of dexamethasone and lactoferrin *in vivo* induced PMN-MDSCs compared to tumor PMN-MDSCs

Fig. S6. GO_cfp classification of dexamethasone and lactoferrin *in vivo* induced PMN-MDSCs compared to tumor PMN-MDSCs

Fig. S7. Western Blot of LRP1 in PMN-MDSCs induced by PBS (Con) or DXM/lactoferrin *in vitro* from mouse BM cells and human PBMCs. (n = 3 biologically independent samples).

Fig. S8. CFSE labelling efficiency of Con PMN-MDSCs, DXM/lactoferrin (D/L) PMN-MDSCs, and tumor PMN-MDSCs.

Fig. S9. Transferred/host PMN-MDSC ratio in spleen, PBMCs and tumor 1 (a), 2 (b) and 3 (c) days after i.v. injection of CFSE labeled PBS (Con) or DXM/lactoferrin (D/L) *in vitro* from mouse BM cells and PMN-MDSC from B16 tumor bearing mice into tumor bearing recipients. Typical example of flow cytometric analysis is shown.

Fig. S10. IFN- γ ⁺ cells in CD8 T cells from PBMC (a) and tumor (b) of B16 tumor bearing mice 24 hours after i.v. injection of Con PMN-MDSCs, DXM/lactoferrin (D/L) PMN-MDSCs, and tumor PMN-MDSCs. PBS was used as vehicle control. Typical example of flow cytometric analysis is shown.

Table. S1. REAGENT

Name	Company
RPMI 1640	Invitrogen
CFSE	Invitrogen
LF (lactoferrin)	Sigma-Aldrich
TRIzol reagent	Invitrogen
ficoll	GE
Recombinant murine GM-CSF	R & D
Recombinant murine IL6	R & D
Recombinant human GM-CSF	R & D
Recombinant human IL6	R & D
LRP1 Antibody (A2MR-beta-1)	Invitrogen
HPGDS Polyclonal Antibody	Invitrogen
PTGS2 Polyclonal Antibody	Invitrogen
Anti-beta Actin (ab8226)	Abcam

Table. S2. antibodies used for flow cytometer and western blot analysis

REAGENT or RESOURCE	SOURCE	IDENTIFIER
Antibodies	Company	Clone
Anti-mouse CD3 functional grade	eBioscience	16-0031-86
Anti-mouse CD8 functional grade	eBioscience	16-0281
Anti-mouse CD11b-FITC/PerCP-Cy5.5	eBioscience	M1/70.15
Anti-mouse Ly6C-PE	eBioscience	HK1.4
Anti-mouse Ly6G- PerCP-cy5.5	BD Biosciences	1A8
Anti-mouse CD4-PE-cy7	eBioscience	GK1.5
Anti-mouse CD8-APC	eBioscience	53-6.7
Anti-mouse CD3-PE	eBioscience	12-0031
Anti-mouse CD19-eFluor 450	eBioscience	1D3
Anti-mouse CCR3-APC	Biolegend	J073E5
Anti-mouse IFN- γ -FITC	Biolegend	XMG1.2
Anti-mouse Gr-1-PE-Cy7	eBioscience	RB6-5931
Anti-human CD3 functional grade	eBioscience	16-0039-81
Anti-human CD8 functional grade	eBioscience	16-0289-81
Anti-human CD11b-FITC/PerCP-Cy5.5	eBioscience	M1/70
Anti-human HLA-DR-APC	eBioscience	LN3
Anti-human CD14-PE-cy7	eBioscience	61D3
Anti-human CD15-PB	eBioscience	HI98
Anti-human CD4-PE	eBioscience	MHCD0404

Anti-human CD8-PE-Cy7

eBioscience

RPA-T8

Table. 3. Primers used for qRT-PCR

Ms-Lrp1-for	5'- ACTTCTCGGATGCCACCTTG-3'
Ms-Lrp1-rev	5'- TGCTGCAGTCTGTGGTACTG-3'
Ms-Lrp2-for	5'- GGCAGTGGGAATTTTCGCTG-3'
Ms-Lrp2-rev	5'- CAGGAGCTAGGGATGCAGG-3'
Ms-Ptgs1-for	5'- ATGAGTCGAAGGAGTCTCTCG -3'
Ms-Ptgs1-rev	5'- GCACGGATAGTAACAACAGGGA -3'
Ms-Ptgs2-for	5'- TTCCAATCCATGTCAAACCGT -3'
Ms-Ptgs2-rev	5'- AGTCCGGGTACAGTCACACTT -3'
Ms-hpgds-for	5'- AAGCTGACTGGCCTAAAATCAAG -3'
Ms-hpgds-rev	5'- CTCTGGTGGATTGTAAGTCCTTC -3'
Ms-hpgd-for	5'-CAGGGCATAGGCAAAGCCTT -3'
Ms-hpgd-rev	5'-ACGCCTGCATTGTTGACCAA -3'
Ms-Itln1-for	5'-TGACAATGGTCCAGCATTACC-3'
Ms-Itln1-rev	5'-ACGGGGTTACCTTCTGGGA-3'
Ms-β-actin-for	5'-CGTGCGTGACATCAAAGAGAAG-3'
Ms-β-actin-rev	5'-CGTTGCCAATAGTGATGACCTG-3'
Hu-β-actin-for	5'-CTCCATCCTGGCCTCGCTGT-3'
Hu-β-actin-rev	5'-GCTGTACCTTCACCGTTCC-3'
Hu-Ptgs1 for	5'- CGCCAGTGAATCCCTGTTGTT -3'
Hu-Ptgs1 rev	5'- AAGGTGGCATTGACAAACTCC -3'
Hu-Ptgs2-for	5'-CTGGCGCTCAGCCATACAG -3'
Hu-Ptgs2 rev	5'-CGCACTTATACTGGTCAAATCCC -3'
Hu-hpgds-for	5'- ACCAGAGCCTAGCAATAGCAA -3'

Hu-hpgds rev	5'- AGAGTGTCCACAATAGCATCAAC -3'
Hu-hpgd-for	5'-CAGCGTTGGCTGCTAATCTTA-3'
Hu-hpgd-rev	5'-AGCCTGGACAAATGGCATTCA-3'
Hu-Lrp1-for	5'- CTATCGACGCCCCTAAGACTT-3'
Hu-Lrp1-rev	5'- CATCGCTGGGCCTTACTCT-3'
Hu-ITLN1-for	5'-ACGTGCCCAATAAGTCCCC-3'
Hu-ITLN1-rev	5'-CCGTTGTCAGTCCAACACTTTC-3'